# Safety Profile and Lack of Immunogenicity of IncobotulinumtoxinA in Pediatric Spasticity and Sialorrhea: A Pooled Analysis

**DOI:** 10.3390/toxins14090585

**Published:** 2022-08-25

**Authors:** Steffen Berweck, Marta Banach, Deborah Gaebler-Spira, Henry G. Chambers, A. S. Schroeder, Thorin L. Geister, Michael Althaus, Angelika Hanschmann, Matteo Vacchelli, Michaela V. Bonfert, Florian Heinen, Edward Dabrowski

**Affiliations:** 1Schön Klinik Vogtareuth, Krankenhausstraße 20, 83569 Vogtareuth, Germany; 2Dr. von Hauner Children’s Hospital, Munich University, Lindwurmstrasse 4, 80337 Munich, Germany; 3Jagiellonian University Medical College, Świętej Anny 12, 31-008 Krakow, Poland; 4Shirley Ryan AbilityLab, Northwestern Feinberg School of Medicine, 355 E. Erie St, Chicago, IL 60611, USA; 5Rady Children’s Hospital, 3030 Children’s Way MC 5062, San Diego, CA 92123, USA; 6Division of Paediatric Neurology and Developmental Medicine, LMU Center for Children with Medical Complexity, Dr. von Hauner Children’s Hospital, Ludwig Maximilian University of Munich, Lindwurmstrasse 4, 80337 Munich, Germany; 7Merz Therapeutics GmbH, Eckenheimer Landstraße 100, 60318 Frankfurt am Main, Germany; 8Beaumont Pediatric Physical Medicine & Rehabilitation–Royal Oak, 3535 W 13 Mile Rd #307, Royal Oak, MI 48073, USA

**Keywords:** botulinum toxin, incobotulinumtoxinA, muscle spasticity, all movement disorders, all pediatric, sialorrhea, safety, immunogenicity, antibodies

## Abstract

IncobotulinumtoxinA, a pure botulinumtoxinA formulation, is free of accessory proteins. This analysis provides pooled safety data from phase 3 trials of children/adolescents (2–17 years), investigating incobotulinumtoxinA for the treatment of spasticity associated with cerebral palsy (at doses ≤20 U/kg (max. 500 U) per injection cycle (IC) for ≤6 ICs; three trials) or sialorrhea associated with neurologic disorders (at total doses of 20–75 U per IC for ≤4 ICs; one trial) for ≤96 weeks. Safety endpoints included the incidences of different types of treatment-emergent adverse events (TEAEs) and immunogenicity. IncobotulinumtoxinA dose groups were combined. Of 1159 patients (mean age 7.3 years, 60.4% males) treated with incobotulinumtoxinA, 3.9% experienced treatment-related TEAEs, with the most common being injection site reactions (1.3%) (both indications), muscular weakness (0.7%) (spasticity), and dysphagia (0.2%) (sialorrhea). Two patients (0.2%) experienced a treatment-related treatment-emergent serious adverse event, and 0.3% discontinued the study due to treatment-related TEAEs. No botulinumtoxinA-naïve patients developed neutralizing antibodies (NAbs) after incobotulinumtoxinA. All children/adolescents with known pre-treatment status and testing positive for Nabs at final visit (*n* = 7) were previously treated with a botulinumtoxinA other than incobotulinumtoxinA. IncobotulinumtoxinA was shown to be safe, with very few treatment-related TEAEs in a large, diverse cohort of children/adolescents with chronic conditions requiring long-term treatment and was without new NAb formation in treatment-naïve patients.

## 1. Introduction

BotulinumtoxinA (BoNT-A) is used to treat several conditions affecting children/adolescents, such as spasticity, chronic sialorrhea (drooling), overactive bladder, and strabismus [1,2,3,4,5]. Since many of these conditions require repeated and long-term interdisciplinary therapy (including with BoNT-A) [2,3,6], which may need careful adjustment over time to fit the patient’s development and needs [7,8], both safety and long-term treatment response are paramount considerations when treating pediatric populations with BoNT-A.

As with other biologic drugs, long-term treatment with BoNT-A can be associated with immunogenicity that limits the usefulness of established treatments, may lead to higher required treatment doses, or can be the reason for secondary nonresponse to treatment [6,9,10]. However, not all BoNT-A formulations have the same potential for immunogenicity [6,11,12,13]. The relative immunogenicity of each BoNT formulation is determined by both the amount of core neurotoxin protein (the active component) and the amount of accessory clostridial proteins (the nontherapeutic component) [6,10]. Although only neutralizing antibodies (Nabs) against the core neurotoxin can interfere with the therapeutic effect and result in clinical nonresponsiveness [14], activation of the immune response by the accessory proteins, particularly the hemagglutinin-1 protein, can facilitate this unwanted NAb development [10,13,15]. Thus, one approach to reducing the risk of developing NAbs is to choose a BoNT-A formulation that contains only active neurotoxin proteins with no accessory protein content [6,10,12].

IncobotulinumtoxinA is an approved, highly purified BoNT-A formulation containing only the active neurotoxin [16]. While other BoNT-As on the market contain varying amounts of accessory proteins, all accessory proteins and other bacterial substances, such as flagellin, are removed from incobotulinumtoxinA during manufacturing [6,10]. These reasons are thought to explain the low risk of immunogenicity seen with incobotulinumtoxinA [6,11].

In the current report, pooled analyses of safety and immunogenicity data from four phase 3 clinical trials investigating incobotulinumtoxinA for the treatment of lower limb (LL) and/or upper limb (UL) spasticity associated with cerebral palsy (CP) (Treatment with IncobotulinumtoxinA in Movement (TIM) [17]; Treatment with IncobotulinumtoxinA in Movement Open-label (TIMO) [18]; incobotulinumtoXinA in aRm treatment in cerebral pAlsy (XARA) [19]) or sialorrhea associated with neurological disorders (Sialorrhea Pediatric Xeomin Investigation (SIPEXI) [20]) in children/adolescents (aged 2–17 years) are presented. This large database, which includes findings from more than 1100 children/adolescents treated for up to 96 weeks, provides insights into both the safety and immunogenicity profiles of incobotulinumtoxinA.

## 2. Results

### 2.1. Demographics

This pooled trial analysis included 1159 patients who received at least one injection cycle (IC) of incobotulinumtoxinA for either spasticity (*n* = 907) or sialorrhea (*n* = 252). The overall mean age was 7.3 years, and 60.4% were males. Overall, 44.3% of patients were ambulatory without the need for a hand-held walking device (Gross Motor Function Classification System [GMFCS] levels I–II [21]), and 28.3% had severe motor impairments (GMFCS levels IV–V [21]) (Table 1).

The mean number of ICs was 3.6 for both patients treated for spasticity (maximum 6 ICs) or sialorrhea (maximum 4 ICs). Overall, 975 (84.1%) children/adolescents completed their respective trial (753 (83.0%) treated for spasticity and 222 (88.1%) for sialorrhea).

### 2.2. Safety: Pediatric Spasticity and Sialorrhea

At least one treatment-emergent adverse event (TEAE) was reported at any time during repeated cycles of incobotulinumtoxinA for 481 of 1159 (41.5%) children. A total of 45 (3.9%) participants experienced events considered treatment-related by investigators, all but one of which were mild or moderate in severity. A total of 23 participants (2.0%) experienced at least one TEAE of special interest (TEAESI), potentially indicative of toxin spread, the vast majority of whom experienced events that were mild to moderate in intensity (*n* = 22). A total of 15 (1.3%) TEAESIs were considered treatment-related by the respective investigator (Table 2).

Overall, 58 (5.0%) children/adolescents reported treatment-emergent serious adverse events (TESAEs), two of whom had events considered treatment-related by investigators; both patients were being treated for spasticity (Table 2). There was no difference in the frequency of TESAEs by age: 5.1% of 922 children aged 2–11 years developed a TESAE versus 11 (4.6%) of 237 children aged 12–17 years. The events were considered treatment-related in 3 (0.3%) of 12 patients with spasticity and 1 (0.4%) of 6 patients with sialorrhea who discontinued treatment because of TEAEs (Table 2). There were no fatal TEAEs.

The proportions of children/adolescents reporting TEAEs with incobotulinumtoxinA by indication (for the treatment of spasticity or sialorrhea; 40.0%, 46.8%, respectively) were similar to those of the overall treated population (41.5%) (Table 2).

### 2.3. TEAEs by IC

Figure 1 presents the overall incidence of TEAEs by IC for patients treated with incobotulinumtoxinA. The incidence of TEAEs per IC was <22% and tended to decrease over subsequent injection cycles, with the highest numbers occurring in IC1 and the fewest occurring in IC5. Treatment-related TEAEs occurred infrequently in each IC (with <2.0% incidence). No treatment-related TEAEs occurred during the fifth and sixth ICs.

The pattern of decreasing TEAE incidence with repeated dosing across all six ICs was seen specifically in children/adolescents treated for spasticity together with a low incidence of treatment-related TEAEs (≤2%) (Figure 2). In children/adolescents treated for sialorrhea, the TEAE and treatment-related TEAEs were reported with similar frequency across all four ICs; again, the incidence of treatment-related TEAEs was low across all ICs (<3.0%; Figure 3).

### 2.4. Most Common TEAEs by Indication

TEAEs reported during repeated incobotulinumtoxinA ICs generally showed some relation to indication. For instance, 13 (1.4%) of those treated for spasticity reported musculoskeletal and connective tissue disorders as treatment-related TEAEs, but none were reported in the group treated for sialorrhea. Gastrointestinal disorders, including dysphagia and dry mouth, were more common in those treated for sialorrhea.

#### 2.4.1. Spasticity

The most common TEAEs reported in children/adolescents with spasticity during up to six ICs of incobotulinumtoxinA were as would be expected in this young population. The most frequent events (in >2.0% of children/adolescents) were nasopharyngitis (8.7%), bronchitis (4.6%), upper respiratory tract infection (3.9%), pharyngitis (3.6%), viral respiratory tract infection (2.8%), pyrexia (2.8%), and pain in the extremity (2.1%), with >95% of each event being mild or moderate in severity.

All treatment-related TEAEs, the most common of which are shown in Table 3, were mild or moderate in severity, except for one case of seizure that was also considered a TESAE and led to discontinuation of incobotulinumtoxinA. The other reported treatment-related TESAE was an influenza-like illness that led to discontinuation of incobotulinumtoxinA. One case of eyelid ptosis, considered treatment-related by the investigator but not serious, led to discontinuation of treatment.

TEAESIs reported during up to six incobotulinumtoxinA ICs in children/adolescents with spasticity were most commonly (in ≈2.0% of children/adolescents) muscular weakness (*n* = 6; 0.7%) (Table 3), dyspnea, constipation, and dysphagia (*n* = 3; 0.3%), respectively.

#### 2.4.2. Sialorrhea

TEAEs in this patient group again reflected their age, with nasopharyngitis (7.5%), pharyngitis (7.5%), pyrexia (4.8%), respiratory tract infection (4.4%), headache (3.6%), upper respiratory tract infection (3.2%), cough (3.2%), respiratory disorder (2.8%), viral respiratory tract infection (2.4%), viral upper respiratory tract infection (2.4%), and viral infection (2.4%) being most frequently reported in >2.0% of participants. All these most common TEAEs were mild or moderate in severity.

Only two treatment-related TEAEs were reported in two or more children/adolescents receiving incobotulinumtoxinA for sialorrhea (dysphagia, *n* = 5 and dry mouth, *n* = 2). All treatment-related TEAEs were mild or moderate in severity,, and none were considered serious. Dysphagia, altered saliva, and choking, all considered treatment-related, led to discontinuation of treatment by one adolescent.

The only TEAESI reported during up to four incobotulinumtoxinA ICs in patients with sialorrhea was dysphagia (*n* = 5; 2.0%).

### 2.5. Immunogenicity

Overall, 520 patients treated with incobotulinumtoxinA were eligible for antibody testing at screening, but, ultimately, fluorescence immunoassay (FIA) testing was achievable in 423 (81.3%). At the last visit, 613 patients treated with incobotulinumtoxinA were eligible for antibody testing, and FIA results were achievable for 393 (64.1%) (Table 4). Overall, there were patients for whom no hemidiaphragm assay (HDA) tests were performed, either at screening (*n* = 25) or the last visit (*n* = 22), after a positive FIA result. These HDA results were set to “missing” and were recorded as deviations from the trial protocol. BoNT pre-treatment status was routinely assessed only in trials of patients with spasticity.

Across the four pediatric trials, 60 of 423 tested (14.2%) children/adolescents had a positive FIA test at screening and 54 of 393 tested (13.7%) were FIA-positive at the last visit (Table 4). Using the HDA, 10 of 423 (2.4%) children/adolescents tested positive for NAbs at screening and 10 of 393 (2.5%) were NAb-positive at the last visit (Table 4). Analysis of the subpopulation with known pre-treatment status showed that no BoNT-naïve patients developed NAbs after incobotulinumtoxinA injections; all children with positive NAb tests had previously been treated with a BoNT other than incobotulinumtoxinA (Table 5).

Overall, no cases of secondary nonresponse due to NAb formation were identified in the course of the four trials. Most subjects with positive HDA or FIA results, without further determination of NAbs, as assessed at any time during the study, responded to treatment based on the investigator’s Global Impression of Change of Plantar Flexor Spasticity Scale assessment. One subject with spasticity in the TIM trial, who was HDA-positive at screening but negative at the end of the trial, showed limited or no change in function. In SIPEXI, of the three subjects with positive results in the HDA test (all were positive at screening and at last visit), two showed a clear response to treatment with incobotulinumtoxinA despite the presence of antibodies against BoNT-A. The other subject had received a placebo in the main period (MP) and discontinued the trial at the end of the second IC due to withdrawal by the subject. In this subject, the response to treatment was on the same level for both placebo and incobotulinumtoxinA.

## 3. Discussion

This large pediatric CP population encompassed very young children (2–5 years old) as well as those aged up to 17 years and patients with all levels of motor impairment, including those with severe motor impairments (GMFCS levels IV–V). In addition, these study data are closer to replicating the real-world use of botulinumtoxin in children than are previous clinical trial data. Children/adolescents received consecutive incobotulinumtoxinA injection treatments for spasticity and sialorrhea for a mean of 3.6 ICs and for a maximum period of 96 weeks. IncobotulinumtoxinA was well tolerated, as indicated by a high patient retention rate (overall, 84.1% of the children/adolescents completed their trial, across all trials). Results of the current analyses indicate that, over the utilized multiple ICs, incobotulinumtoxinA has a favorable safety and tolerability profile when administered repeatedly to children/adolescents for spasticity (at doses of up to 20 U/kg (maximum 500 U) per IC for up to six ICs) or sialorrhea (at total doses of 20–75 U per IC for up to four ICs). Only 3.9% (*n* = 45) of 1159 children/adolescents treated with repeated incobotulinumtoxinA ICs developed a treatment-related TEAE. Treatment-related TEAESIs, TESAEs, and TEAEs, leading to discontinuation, occurred infrequently. The findings indicate that the frequency of TEAEs and treatment-related TEAEs generally decreased with repeated injections. Most TEAEs were considered mild or moderate in intensity, and the most common were, as would be expected in predominantly young children, respiratory disorders and pyrexia.

Treatment-related TEAEs were generally associated with the indication (muscle weakness for spasticity and dysphagia for sialorrhea) or route of administration (injection site reactions). Two patients experienced a TESAE, which was considered by the respective local investigator to be treatment-related: one was a seizure and the other was a flu-like illness; both led to discontinuation of treatment.

Seizure was also reported with onabotulinumtoxinA in a study of 381 children with LL spasticity; in this study, 4 (1.6%) of the 254 patients treated with onabotulinumtoxinA had a seizure [22]. These authors noted that seizure is a common CP comorbidity and that many of their patients (19%) had a medical history of seizure, including the four mentioned here. Indeed, Szpindel and colleagues [23] found an elevated risk of epilepsy in two cohorts of children with CP (42.1% using registry data (*n* = 302) and 43.2% using administrative data (*n* = 370) compared to 1.39% in controls (*n* = 6040)), with the prevalence of epilepsy doubling in those with the highest versus the lowest GMFCS level of disability. It should be noted here that the clinical trials described in the current report included participants at all GMFCS levels (I–V), in whom an additional two generalized tonic–clonic seizures (0.2%) and one report of epilepsy were identified but not considered to be treatment-related.

The current findings represent the largest prospectively collected database available to date on the use of incobotulinumtoxinA in children and confirm the findings of a smaller real-world study that found this BoNT to have a good safety profile in pediatric patients [24]. Studies in adults have reported that incobotulinumtoxinA is associated with few treatment-related adverse events (AEs) when administered for a variety of indications, including cervical dystonia [25], spasticity [26], blepharospasm and other neurologic conditions [27], and aesthetic indications [28].

Antibody testing is an important assessment in studies of BoNT-A, although it is burdensome due to the additional required blood samples from patients and, therefore, was only allowed in all incobotulinumtoxinA trials for patients with higher body weights (≥21 kg in TIM, TIMO, and XARA and ≥30 kg in SIPEXI). Overall, the rates of HDA positivity were similar in those with available data prior to treatment (at screening) compared with those with available data after incobotulinumtoxinA treatment (at the last study visit) (2.4% vs. 2.5%). Importantly, among 318 patients with spasticity, known BoNT pre-treatment status, and available antibody testing results, no BoNT-naïve patients developed new NAbs during treatment with incobotulinumtoxinA (i.e., tested negative at the screening visit but tested positive at the individual’s final visit). The seven children with spasticity who had positive NAb tests at screening and the seven with positive results at the final visit had all been pre-treated with a BoNT other than incobotulinumtoxinA. Of clinical importance, pre-existing NAbs did not prevent successful treatment with incobotulinumtoxinA, and no cases of secondary nonresponse due to NAb formation were identified during any of the four trials included in this analysis. In patients with sialorrhea, all three who tested positive at the individual’s final visit had also tested positive at the screening visit; in these cases, whether the patients had been pre-treated with BoNT-A could not be verified. Additionally, information necessary for final determination of the cause of NAb was incomplete for the four pre-treated patients with spasticity who had missing screening results.

There are some data available about the immunogenicity of other BoNT-A formulations in children/adolescents. In a prospective, open-label clinical trial of 207 children with CP-related spasticity treated with onabotulinumtoxinA (average treatment duration 1.46 years), 28% had newly developed NAbs (using the mouse protection assay; MPA) and 6% had detectable NAbs with clinical nonresponse [29]. In a small study of children aged 2 to 6 years with CP-related spasticity, Oshima and colleagues [30] reported a 0% rate of blocking-antibody formation using the MPA after 26 months of treatment with onabotulinumtoxinA administered once a year (*n* = 20), but the rate increased to 11% in those treated three times a year (*n* = 18). Immunogenicity data have also been reported for abobotulinumtoxinA. In a prospective, repeat-treatment, open-label extension (OLEX) of a double-blind study, 207 children aged 2–17 years were treated with abobotulinumtoxinA for CP-related LL spasticity (maximum total dose per IC: 30 U/kg or 1000 U). After up to four injections of abobotulinumtoxinA over the course of 1 year, NAbs were detected in 2.1% (4/193) of patients; all those positive for NAbs had received prior treatment with BoNT-A, although the prior formulation and method for detecting NAbs were not specified in the report [31]. Delgado and colleagues [32] reported a similar rate of NAb formation (2.3%) using the MPA in children with CP treated with abobotulinumtoxinA (8 U/kg or 16 U/kg) for UL spasticity for up to four ICs; all had been pre-treated with BoNT-A. It should be noted that the MPA is a considerably less sensitive assay than the HDA used in the current study [33]. For clinicians, these reports highlight the importance of considering the immunologic differences in accessory protein content among BoNT-A formulations, as the presence of NAbs can result in secondary nonresponse to treatment [34].

Most insights with regard to the potential for immunogenicity and risk of secondary nonresponse with BoNTs have come from the longer established adult indications. The recent evidence regarding BoNT-A treatment in adults does indeed indicate that the risk of provoking an immune response differs among BoNT-A formulations [11], with higher levels of antibodies produced by BoNT-A formulations with higher protein loads [6,9,12]. Preclinical and clinical studies data suggest that incobotulinumtoxinA carries less risk of inducing an immunogenic response relative to other BoNT-As because of its high degree of purity, reflecting the removal of BoNT-A accessory proteins [35,36]. For example, a cross-sectional analysis of antibody prevalence across indications and BoNT-A formulations found NAb prevalence was similar for abobotulinumtoxinA and onabotulinumtoxinA but absent with incobotulinumtoxinA [14]. A meta-analysis of 43 studies involving 8833 patients over the period 2000–2020 reported that abobotulinumtoxinA produced the highest incidence of NAbs (7.4%) versus <0.003% for incobotulinumtoxinA and onabotulinumtoxinA (*p* < 0.003) [37].

While many studies have found BoNT-A treatment to be safe in children [38], not all BoNT-A formulations are the same, and, in fact, they may be considered distinct medications. While the 150 kDa (ng protein/100 U) neurotoxin molecules in each formulation are equally active, different formulations contain different quantities of neurotoxin. Units of BoNT-A for each formulation are not necessarily bioequivalent, influencing effectiveness, dosing, and safety, and thus should not be treated as interchangeable [13]. An emerging body of evidence emphasizes the dual goals of minimizing the immune response and maximizing patient responsiveness when treating a child/adolescent with BoNT-A [6]. In response, both the United States Food and Drug Administration [39] and the European Medicines Agency [40] now recommend evaluating and mitigating adverse immunologically related responses associated with therapeutic protein products and support taking actions to reduce risk. Choosing a BoNT-A formulation that offers a high degree of purity can be one of those steps and should be considered as first-line BoNT-A therapy to avoid possible future treatment failure [11,41].

### Strengths and Weaknesses

The strength of this report is the availability of safety and antibody data from a large population of children/adolescents treated with incobotulinumtoxinA for two different indications. The participants represented a broad age range (2–17 years) and all levels of spasticity disease severity (GMFCS-Expanded and Revised levels I–V). Participants received an average of 3.6 ICs of incobotulinumtoxinA (with a maximum of four ICs for those treated for sialorrhea and six for those treated for spasticity) and were followed for up to 96 weeks, with 84.1% completing the trials. Although the investigated exposure time of up to 96 weeks is similar to that of other research (12 to 26 months [29,30,31,32]), real-world use beyond this time is something that warrants further research for this chronic condition.

The lack of a placebo control in one of the sialorrhea cohorts and all the spasticity trials may be viewed as a limitation. However, the administration of one or more placebo injections to very young children (2–5 years old) was considered unethical. Similarly, antibody data were not available for some children, reflecting in part the weight restrictions placed to make sure the participants could tolerate the blood sampling procedure; even so, sampling was not always practical for all eligible patients, such that, for some, the volume obtained was too low for analysis. In addition, information regarding BoNT pre-treatment was not available for all patients.

Another potential limitation was the pooling of incobotulinumtoxinA doses. The decision to pool dose groups was made for two reasons. First, we were able to maximize the sample size by pooling the data, which was feasible because the trials shared many enrollment criteria. Second, although improvements in spasticity can differ for the investigated dose groups [17], useful levels of improvement in spasticity can also be achieved with lower dose regimens [17,18]. By pooling data from each dose group, larger sets of TEAEs could be analyzed, allowing the interpretation to be closer to the real-world situation of clinical decision-making when caring for children with CP than has previously been seen in clinical trials.

## 4. Conclusions

Choosing a BoNT-A with a good safety and immunogenicity profile is a critical consideration when treating children/adolescents with chronic conditions requiring long-term therapy. The results of the current analysis show that incobotulinumtoxinA, given in as many as six ICs for up to 96 weeks, was associated with very few treatment-related TEAEs in more than 1100 children/adolescents representing a broad range of age and disability levels. Moreover, these data confirm that incobotulinumtoxinA in treatment-naïve patients lacks an immunologic response and does not induce new NAb formation during extended use. These findings in children/adolescents add to the growing body of evidence from studies in adults that the highly purified composition of incobotulinumtoxinA makes it a preferred choice for chronic conditions that require BoNT treatment.

## 5. Materials and Methods

### 5.1. Participants

All participants were enrolled in one of four phase 3 clinical trials of incobotulinumtoxinA. The TIM, TIMO, and XARA trials enrolled children/adolescents 2–17 years of age with unilateral or bilateral LL and/or UL spasticity associated with CP. Participants had an Ashworth Scale [42] score of ≥2 at baseline in clinical patterns for treatment and GMFCS [21] scores of I–V. In these trials, patients were excluded if they had previously received BoNT treatment within ≤14 weeks before screening or during the screening period.

The fourth phase 3 clinical trial was SIPEXI, in which the main cohort consisted of participants aged 6–17 years with chronic sialorrhea (≥3 months prior to screening) associated with neurologic disorders (e.g., CP or traumatic brain injury) and/or intellectual disability. A second cohort consisted of a younger age group (2–5 years) with sialorrhea. All patients had severe drooling, as determined by the investigator’s Modified Teacher’s Drooling Scale (rating ≥ 6) [43]. Patients were required to have no prior BoNT treatment for any body region in the year preceding screening or within the screening period and no clinically relevant concurrent conditions.

### 5.2. Trial Designs and Treatment

Full details of the designs and methodologies of the four trials have been published [17,18,19,20] and are summarized in Figure 4.

#### 5.2.1. Spasticity

The three phase 3 trials that investigated incobotulinumtoxinA for the treatment of spasticity in children/adolescents (aged 2–17 years) with CP (Figure 4) were TIM, TIMO, and XARA.

TIM (NCT01893411) was a randomized, double-blind, parallel-group trial in pediatric LL spasticity treatment. Patients received total body incobotulinumtoxinA doses of up to 4–16 U/kg body weight (maximum 100–400 U) per cycle in two controlled ICs every 12–36 weeks. Two clinical patterns were treated: bilateral or unilateral pes equinus was mandatory; if unilateral, the second pattern was ipsilateral flexed knee or adducted thigh [12].

TIMO (NCT01905683) was an open-label, long-term trial of children/adolescents with LL or combined LL and UL spasticity. TIMO included eligible patients who completed TIM and newly recruited patients. Patients received four ICs, each with total body incobotulinumtoxinA doses of up to 16–20 U/kg body weight (maximum 400–500 U), every 12–16 weeks. Two clinical patterns were treated: pes equinus was mandatory, with flexed knee or adducted thigh as options for ipsilateral treatment and/or ULs for unilateral/bilateral treatment [18].

XARA (NCT02002884) was a randomized phase 3 trial of children/adolescents with unilateral or bilateral spasticity (UL or combined UL/LL spasticity) that had a double-blind MP and an OLEX. In the MP, patients received four ICs of incobotulinumtoxinA into the UL(s) of up to 2–16 U/kg body weight (maximum 50–400 U) every 12–16 weeks, with optional LL injections to a maximum total body dose of 20 U/kg body weight (500 U). In the OLEX, patients received three further ICs at the highest MP dose (total body dose 8–20 U/kg body weight, maximum 200–500 U based on dose group and baseline GMFCS score) [19]. At least one primary clinical pattern (flexed wrist and/or flexed elbow) was treated, with the option of three additional possible UL patterns (clenched fist, thumb in palm, pronated forearm), and if clinically required, LL unilateral/bilateral injections could be added in four possible clinical patterns (adducted thigh, flexed knee, pes equinus, or extended great toe).

#### 5.2.2. Sialorrhea

SIPEXI (NCT02270736) was a prospective, multicenter, phase 3 trial in children/adolescents (aged 2─17 years) with chronic sialorrhea due to neurologic disorders (CP [65%], traumatic brain injury [4%], other [31%]), and/or intellectual disability (89%) [20]. Patients received bodyweight-dependent doses of incobotulinumtoxinA (20–75 U) administered in a 3:2 ratio into all submandibular and parotid glands. An MP with one IC (placebo-controlled, double-blind in those aged 6–17 years) was followed by an OLEX with up to three further ICs. An additional cohort of participants aged 2–5 years received active treatment throughout the trial. The trial lasted 72 weeks, with a 16-week follow-up period after each injection.

### 5.3. Standard Protocol Approvals, Registrations, and Patient Consent

All four trials were conducted in accordance with Good Clinical Practice and the ethical principles of the Declaration of Helsinki and registered on clinicaltrials.gov (NCT01893411, NCT01905683, NCT02002884, NCT02270736). The trial protocol, informed consent forms, and other appropriate trial-related documents were reviewed and approved by the local independent ethics committees and institutional review boards. Parent(s)/guardian(s) of all patients provided written informed consent, and patients provided assent (if applicable).

### 5.4. Assessments

#### 5.4.1. Safety

Safety data were pooled across all trials, by indication, and by incobotulinumtoxinA dose. The safety endpoints were the proportions of patients with TEAEs, TEAESI (those events potentially indicative of toxin spread), TESAEs and TEAEs leading to discontinuation from the trial, and events that were treatment-related. TEAEs were assessed for all subjects at each trial visit.

Data from all consecutive ICs were analyzed from the TIM, TIMO, and XARA trials. TIMO data were analyzed together with TIM data, namely, for subjects already treated in TIM and continued in TIMO, the last incobotulinumtoxinA IC with TIM was x and the first in TIMO was x + 1. For newly recruited subjects in TIMO, the first incobotulinumtoxinA IC in TIMO was injection 1.

For SIPEXI, which was placebo-controlled, data from all ICs were analyzed. The first injection in the incobotulinumtoxinA arm of the MP was analyzed as the first IC. For subjects who received placebos in the MP and received their first treatment with incobotulinumtoxinA during the second IC of the trial (i.e., first OLEX injection), this injection was considered the first injection and was analyzed together with data from the incobotulinumtoxinA arm from the MP. Note, subjects who discontinued after the first incobotulinumtoxinA injection in the MP and did not participate in the OLEX of the trial were included in the analysis.

#### 5.4.2. Antibody Measurements

Immunogenicity assessments were conducted only for participants weighing ≥21 kg in TIM, TIMO, and XARA due to the comparatively large blood volume needed. For SIPEXI, immunogenicity assessments were performed only in children weighing ≥30 kg. These weight thresholds were implemented to reduce the burden for younger patients from additional blood sample collection in addition to the mandatory standard laboratory samples.

Antibody samples were always collected before the first treatment injection at the screening visit (spasticity trials) or baseline visit (SIPEXI) and at the final individual visit of each trial. As an exception, for those patients who enrolled in TIM and who progressed to TIMO, the first collection was performed at the TIM screening visit and the second one at the TIMO final visit, usually after up to six ICs of the two trials combined.

Blood samples for immunogenicity testing were screened in a first step using an FIA to detect any antibodies against BoNT. In case of a positive FIA finding, further testing with the highly sensitive mouse ex vivo HDA for the final confirmation of NAb presence and respective determination of titers was performed as a second step.

#### 5.4.3. Statistics

The primary objective of this trial was to report on the safety and lack of immunogenicity of repeated incobotulinumtoxinA injections in children/adolescents. As such, only standard descriptive analyses were performed. Continuous variables for the analysis of demographics were summarized by mean and standard deviation. For qualitative variables, absolute and percent frequencies (*n*, %) were displayed. Percentages were calculated using the denominator of subjects in the respective analysis set.

## Figures and Tables

**Figure 1 toxins-14-00585-f001:**
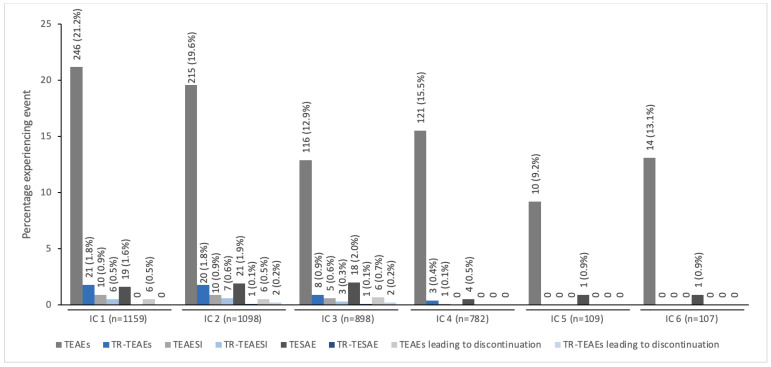
The incidence of TEAEs by IC for pediatric patients with spasticity or sialorrhea treated with incobotulinumtoxinA ^a^. ^a^ All patients with spasticity were enrolled in TIM (Treatment with IncobotulinumtoxinA in Movement) [17], TIMO (Treatment with IncobotulinumtoxinA in Movement Open-label) [18], or XARA (incobotulinumtoXinA in aRm treatment in cerebral pAlsy) [19], and patients with sialorrhea were enrolled in SIPEXI (Sialorrhea Pediatric Xeomin Investigation) [20]. IC, injection cycle; TEAE, treatment-emergent adverse event; TEAESI, treatment-emergent adverse event of special interest (potentially indicative of toxin spread); TESAE, treatment-emergent serious adverse event; TR, treatment-related.

**Figure 2 toxins-14-00585-f002:**
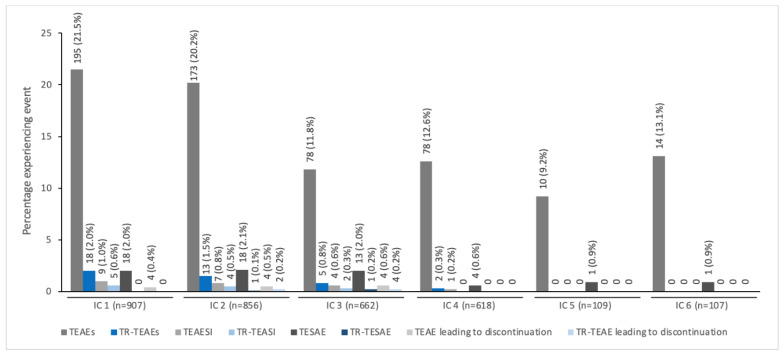
The incidence of TEAEs by IC for pediatric patients with spasticity treated with incobotulinumtoxinA ^a^. ^a^ All patients were enrolled in TIM (Treatment with IncobotulinumtoxinA in Movement) [17], TIMO (Treatment with IncobotulinumtoxinA in Movement Open-label) [18], or XARA (incobotulinumtoXinA in aRm treatment in cerebral pAlsy) [19]. IC, injection cycle; TEAE, treatment-emergent adverse event; TEAESI, treatment-emergent adverse event of special interest (potentially indicative of toxin spread); TESAE, treatment-emergent serious adverse event; TR, treatment-related.

**Figure 3 toxins-14-00585-f003:**
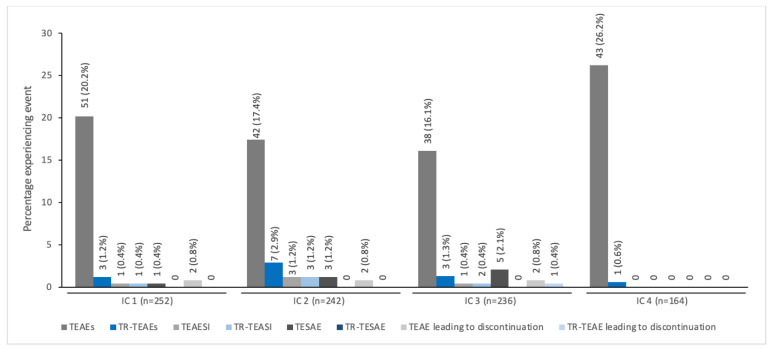
The incidence of TEAEs by IC for pediatric patients with sialorrhea treated with incobotulinumtoxinA ^a^. ^a^ All patients enrolled in SIPEXI (Sialorrhea Pediatric Xeomin Investigation) [20]. IC, injection cycle; TEAE, treatment-emergent adverse event; TEAESI, treatment-emergent adverse event of special interest (potentially indicative of toxin spread); TESAE, treatment-emergent serious adverse event; TR, treatment-related.

**Figure 4 toxins-14-00585-f004:**
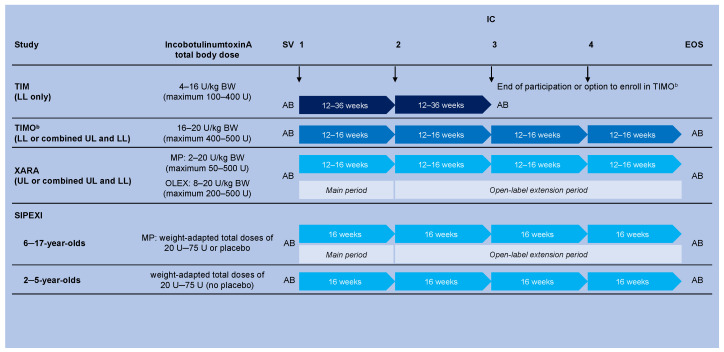
Simplified overview of the study designs of the four phase 3 pediatric studies TIM, TIMO, XARA, and SIPEXI ^a^. ^a^ Full methodologic details can be found in: TIM [17], TIMO [18], XARA [19], and SIPEXI [20]. ^b^ Maximum duration of treatment was 96 weeks for those from the TIM study who entered TIMO. AB: For TIM patients who did not progress to TIMO, AB was performed at the SV and the EOS visit (e.g., after two ICs). For those enrolled in TIM and who progressed to TIMO, AB was performed at the TIM SV and the TIMO EOS visit (e.g., after up to six ICs). For patients newly enrolled in TIMO and for patients in XARA and SIPEXI, AB was performed at the SV and the EOS visit (e.g., after four ICs). AB was restricted to patients weighing ≥21 kg body weight in the spasticity studies and ≥30 kg body weight in the sialorrhea study to reduce the burden of further blood sample requirements on smaller, younger patients. AB, antibody testing; BW, body weight; EOS, end of study; IC, injection cycle; LL, lower limb; MP, main period; OLEX, open-label extension; SIPEXI, Sialorrhea Pediatric Xeomin Investigation; SV, screening visit; TIM, Treatment with IncobotulinumtoxinA in Movement; TIMO, Treatment with IncobotulinumtoxinA in Movement Open-label; U, unit; UL, upper limb; XARA, incobotulinumtoXinA in aRm treatment in cerebral pAlsy.

**Table 1 toxins-14-00585-t001:** Demographics of pediatric patients in phase 3 studies of incobotulinumtoxinA by indication and overall.

Indication	Spasticity ^a^	Sialorrhea ^b^	Overall
*n*	907	252	1159
Male sex, *n* (%)	541 (59.6)	159 (63.1)	700 (60.4)
Age, years, mean (SD)	6.7 (4.2)	9.4 (3.7)	7.3 (4.3)
Weight, kg, mean (SD)	23.3 (13.9)	27.6 (11.8)	24.2 (13.6)
GMFCS-E&R levels ^c^			
I–II	452 (49.8)	61 (24.2)	513 (44.3)
III	206 (22.7)	25 (9.9)	231 (19.9)
IV–V	249 (27.5)	79 (31.3)	328 (28.3)
missing	0 (0.0)	87 (34.5)	87 (7.5)
Number of ICs (Mean, SD)	3.59 (1.31)	3.55 (0.73)	3.58 (1.21)
Pre-treatment with BoNT ^d^ (Yes), *n* (%)	462 (50.9)	Missing ^e^	462 (39.9)
Location of spasticity:			
LL	849 (93.6%)	n/a	n/a
UL	454 (50.1%)	n/a	n/a
Affected body side			
Left	138 (15.2)	n/a	n/a
Right	168 (18.5)	n/a	n/a
Bilateral	601 (66.3)	n/a	n/a

^a^ Patients enrolled in TIM (Treatment with IncobotulinumtoxinA in Movement) [17], TIMO (Treatment with IncobotulinumtoxinA in Movement Open-label) [18], and XARA (incobotulinumtoXinA in aRm treatment in cerebral pAlsy) [19]. ^b^ Patients enrolled in SIPEXI (Sialorrhea Pediatric Xeomin Investigation) [20]. ^c^ GMFCS levels [21] were determined in patients in the spasticity trials and in patients with sialorrhea who had CP. ^d^ Pre-treated was defined as treatment with BoNT for any indication at any time prior to the study. ^e^ Information concerning pre-treatment with any BoNT for any indication was not routinely assessed in the SIPEXI study. BoNT, botulinumtoxin; CP, cerebral palsy; GMFCS-E&R, Gross Motor Function Classification System–Expanded and Revised; ICs, injection cycles; LL, lower limb; n/a, not applicable; SD, standard deviation; UL, upper limb.

**Table 2 toxins-14-00585-t002:** TEAEs by indication and overall in children/adolescents with spasticity or sialorrhea receiving repeated doses of incobotulinumtoxinA.

*n* (%) Subjects with:	Spasticity ^a^ (*n* = 907)	Sialorrhea ^b^ (*n* = 252)	Overall (*n* = 1159)
Any TEAE	363 (40.0)	118 (46.8)	481 (41.5)
TR TEAE	33 (3.6)	12 (4.8)	45 (3.9)
Any TEAESI	18 (2.0)	5 (2.0)	23 (2.0)
TR TEAESI	10 (1.1)	5 (2.0)	15 (1.3)
Any TESAE	49 (5.4)	9 (3.6)	58 (5.0)
TR TESAE	2 (0.2)	0 (0.0)	2 (0.2)
Any TEAE leading to discontinuation	12 (1.3)	6 (2.4)	18 (1.6)
TR TEAE leading to discontinuation	3 (0.3)	1 (0.4)	4 (0.3)
Any fatal TEAE	0 (0.0)	0 (0.0)	0 (0.0)

^a^ Patients enrolled in TIM (Treatment with IncobotulinumtoxinA in Movement) [17], TIMO (Treatment with IncobotulinumtoxinA in Movement Open-label) [18], or XARA (incobotulinumtoXinA in aRm treatment in cerebral pAlsy) [19]. ^b^ Patients enrolled in SIPEXI (Sialorrhea Pediatric Xeomin Investigation) [20]. TEAE, treatment-emergent adverse event; TEAESI, treatment-emergent adverse event of special interest (potentially indicative of toxin spread); TESAE, treatment-emergent serious adverse event; TR, treatment-related.

**Table 3 toxins-14-00585-t003:** Most common treatment-related TEAEs (≥2 subjects) in 907 children/adolescents with spasticity during up to six ICs of incobotulinumtoxinA ^a^.

Event	*n* (%) with Event
Muscular weakness	6 (0.7)
Pain in extremity	4 (0.4)
Myofascial pain syndrome	2 (0.2)
Injection site pain	5 (0.6)
Injection site erythema	3 (0.3)
Influenza-like illness	2 (0.2)
Pyrexia	2 (0.2)

^a^ All patients were enrolled in TIM (Treatment with IncobotulinumtoxinA in Movement) [17], TIMO (Treatment with IncobotulinumtoxinA in Movement Open-label) [18], or XARA (incobotulinumtoXinA in aRm treatment in cerebral pAlsy) [19]. ICs, injection cycles; TEAE, treatment-emergent adverse event.

**Table 4 toxins-14-00585-t004:** Immunogenicity finding in children/adolescents with spasticity or sialorrhea receiving repeated doses of incobotulinumtoxinA, ^a^ overall, and by BoNT pre-treatment status.

	Study Screening				Last Study Visit ^b^			
Indication	N Eligible ^c^	N Tested	Positive FIA Test, *n* (%)	Positive HDA Test, *n* (%) ^d^	N Eligible ^c^	N Tested	Positive FIA Test, *n* (%)	Positive HDA Test, *n* (%) ^e^
Overall	520	423	60 (14.2)	10 (2.4)	613	393	54 (13.7)	10 (2.5) ^f^
Sialorrhea	163	80	10 (12.5)	3 (3.8)	197	75	8 (10.7)	3 (4.0) ^f^
Spasticity	357	343	50 (14.6)	7 (2.0)	416	318	46 (14.5)	7 (2.2)
Treatment-naïve	141	139	13 (9.4)	0 (0.0)	164	125	14 (11.2)	0 (0.0)
Pre-treated ^g^	216	204	37 (18.1)	7 (3.4)	252	193	32 (16.6)	7 (3.6)

^a^ All patients with spasticity were enrolled in TIM (Treatment with IncobotulinumtoxinA in Movement) [17], TIMO (Treatment with IncobotulinumtoxinA in Movement Open-label) [18], or XARA (incobotulinumtoXinA in aRm treatment in cerebral pAlsy) [19], and patients with sialorrhea were enrolled in SIPEXI (Sialorrhea Pediatric Xeomin Investigation) [20]. Only children with a body weight ≥30 kg (SIPEXI) or ≥21 kg (TIM, TIMO, XARA) were eligible for antibody testing due to the comparatively large blood volume needed. ^b^ Last visit refers to an individual’s last visit, meaning that at least one post-baseline antibody measurement was available. ^c^ Number of children/adolescents for whom antibody assessments could have been performed. ^d^ HDA results were missing for 25 children/adolescents overall, 3 with sialorrhea, 22 with spasticity overall, 6 who were BoNT-naïve, and 16 who were pre-treated with positive FIA results. ^e^ HDA results were missing for 22 children/adolescents overall, 0 with sialorrhea, 22 with spasticity, 9 who were BoNT-naïve, and 13 who were pre-treated and had positive FIA results. ^f^ One additional child/adolescent had a borderline HDA result. ^g^ Pre-treatment was defined as pre-treatment with BoNT for any indication at any time prior to study entry. FIA, fluorescence immunoassay; HDA, hemidiaphragm assay.

**Table 5 toxins-14-00585-t005:** Listing of subjects with positive HDA results: TIM, TIMO, XARA, and SIPEXI studies pooled ^a^.

Patient	Indication	Pre-Treated/Naïve	HDA Result at Screening Visit ^b^	HDA Result at Individual Final Visit ^b^
1	Spasticity	Pre-treated	Positive	Positive
2	Spasticity	Pre-treated	Missing	Positive
3	Spasticity	Pre-treated	Positive	Positive
4	Spasticity	Pre-treated	Missing	Positive
5	Spasticity	Pre-treated	Missing	Positive
6	Spasticity	Pre-treated	Positive	Positive
7	Spasticity	Pre-treated	Missing	Positive
8	Spasticity	Pre-treated	Positive	Missing
9	Spasticity	Pre-treated	Positive	Negative
10	Spasticity	Pre-treated	Positive	Missing
11	Spasticity	Pre-treated	Positive	Missing
12	Sialorrhea	Undetermined	Positive	Positive
13	Sialorrhea	Undetermined	Positive	Positive
14	Sialorrhea	Undetermined	Positive	Positive

^a^ All patients with spasticity were enrolled in TIM (Treatment with IncobotulinumtoxinA in Movement) [17], TIMO (Treatment with IncobotulinumtoxinA in Movement Open-label) [18], or XARA (incobotulinumtoXinA in aRm treatment in cerebral pAlsy) [19], and patients with sialorrhea were enrolled in SIPEXI (Sialorrhea Pediatric Xeomin Investigation) [20]. Only children with a body weight ≥30 kg (SIPEXI) or ≥21 kg (TIM, TIMO, XARA) were eligible for antibody testing due to the comparatively large blood volume needed. ^b^ The seven children with spasticity who had positive NAb tests at screening and the seven with positive results at the final visit had all received prior treatment with a BoNT other than incobotulinumtoxinA (four with onabotulinumtoxinA, one with abobotulinumtoxinA, and two with both onabotulinumtoxinA and abobotulinumtoxinA). However, the three children with positive HDA test results in SIPEXI, all of whom were positive both at screening and last visit, had not previously received BoNT for sialorrhea, according to their caregivers. HDA, hemidiaphragm assay.

## Data Availability

No individual de-identified patient data are shared. The data source was the integrated database of Merz-sponsored pediatric studies. Key elements of the study protocols, designs, and statistical analysis plans were deposited in the U.S. National Library of Medicine database (www.clinicaltrials.gov, NCT01893411, NCT01905683, NCT02002884, NCT02270736) and EU Clinical Trials Register (https://eudract.ema.europa.eu/, 2012-005054-30, 2012-005055-17, 2012-005496-14, 2013-004532-30). All relevant information is contained within this manuscript.

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
