# Peer review of "Safety Profile and Lack of Immunogenicity of IncobotulinumtoxinA in Pediatric Spasticity and Sialorrhea: A Pooled Analysis"

_toxins, 2022, doi:10.3390/toxins14090585_

Round 1
Reviewer 1 Report
Very good summary paper.
It would be improved if authors addressed a few issues, which while not inaccurate, are somewhat misleading:
While the authors stress that no INCO treatment naive patients developed antibodies, they should note that antibody presence increases over time, and the studies were short compared to real world use. They compare the development of Ab to other toxin formulations in other studies, but some of these were of longer duration, and thus not a direct comparison.
Likewise, while it is accurate that all of the patients that had positive Ab testing at the end of study had been previously exposed to a different formulation of toxin, 4 of the 14 had unknown Ab presence at the start of the study, so that one cannot say with certainty that they did not develop Ab during exposure to INCO. Similarly, the prior exposure to the siallorhea patients was unknown, although the source of their Ab at the start of the study was unknown as well
Author Response
Reviewer 1
Comment: While the authors stress that no INCO treatment naive patients developed antibodies, they should note that antibody presence increases over time, and the studies were short compared to real world use. They compare the development of Ab to other toxin formulations in other studies, but some of these were of longer duration, and thus not a direct comparison.
Response: Thank you for your insightful comment. Looking at the other studies cited in the next paragraph, the duration of the 4 trials analysed in our report (maximum 96 weeks) is not out of line with the other studies mentioned: Koman 2001, 1.46 years; Oshima et al 2017, 26 months; Delgado 2017, 1 year; Delgado 2021, 1 year). Still, you raise an important point that duration was not as long as potential real-world use in this chronic condition. We have included the following text in the Strengths and Weaknesses section Lines 364-366: “Although the investigated exposure time of up to 96 weeks is similar to that of other research (12 to 26 months [29–32]), real world use beyond this time is something that warrants further research in this chronic condition.”
Comment: Likewise, while it is accurate that all of the patients that had positive Ab testing at the end of study had been previously exposed to a different formulation of toxin, 4 of the 14 had unknown Ab presence at the start of the study, so that one cannot say with certainty that they did not develop Ab during exposure to INCO. Similarly, the prior exposure to the siallorhea patients was unknown, although the source of their Ab at the start of the study was unknown as well.
Response: Thank you for your thoughtful comment. Please note that in the sentence that begins on Line 290, we were careful to limit our statement to those patients with spasticity, had known BoNT pre-treatment status and available antibody testing results, because of the issues you raise. Nevertheless, we have added further information to the end of this fifth paragraph (Line 302) of the Discussion: “Additionally, information necessary for final determination of the cause of NAb was incomplete for the four pre-treated patients with spasticity who had missing screening results.”
Reviewer 2 Report
In this manuscript, the authors showed the results of the pooled analyses of safety and immunogenicity data from four phase 3 clinical trials investigating incobotulinumtoxinA for the treatment of lower limb (LL) and/or upper limb (UL) spasticity associated with cerebral palsy (CP) (Treatment with IncobotulinumtoxinA in Movement [TIM], Treatment with IncobotulinumtoxinA in Movement Open-label [TIMO], incobotulinumtoXinA in aRm treatment in cerebral pAlsy [XARA] or sialorrhea associated with neurological disorders (Sialorrhea Pediatric Xeomin Investigation [SIPEXI]) in children/adolescents (aged 2–17 years).
This manuscript is interesting; unfortunately, this manuscript needs substantial improvements and corrections before publishing may be possible.
General points:
Please add a list of abbreviations before References section to your manuscript.
Special points:
Important, this manuscript should be substantially improved, i. e., by substantial references in the field.
Key contribution: please correct this words combination.
Introduction
Lines 33-35: please add multiple references at the end of this sentence.
Lines 40-42: please add multiple references at the end of this sentence.
Lines 42-46: please add multiple references at the end of this sentence.
Lines 50-52: please add multiple references at the end of this sentence.
Results
Lines 69-72: please add appropriate references for each scale.
Lines 145-149: please add appropriate references.
Lines 151-166: please add appropriate references for each sentence.
Table 3: please add a appropriate references for each result.
Discussion
Lines 246-258: please add appropriate references for each sentence.
Lines 283-300: please add appropriate references for each sentence.
Materials and Methods
Lines 393-400: please add appropriate references for each scale.
Statistics
Please describe this section very exactly.
Author Response
Reviewer 2
General points:
Please add a list of abbreviations before References section to your manuscript.
Response: A list of abbreviations has been created (see attached) although it does not appear that one is required according to the Toxins journal standards.
Special points:
Important, this manuscript should be substantially improved, i. e., by substantial references in the field.
Response: We have included references, as appropriate, as suggested below.
Key contribution: please correct this words combination
Response: Please clarify the comment: “please correct this words combination”. We are unable to find any inappropriate word combinations.
Introduction
Reviewer: Lines 33-35: please add multiple references at the end of this sentence.
Response: Three specific references that state the need for repeated injections for spasticity [2,3] and sialorrhea and lifelong-treatment [7] have been added (see Lines 33-35). The second part of the sentence, referring to the importance of safety and treatment response, are the authors’ opinion and should not necessitate a reference. We would like to modify the sentence so that it now reads: “Since many of these conditions require repeated and long-term interdisciplinary therapy (including with BoNT-A) [2,3,6] which may need careful adjustment over time to fit the patient’s development and needs [7,8], both safety and long-term treatment response are paramount considerations when treating pediatric populations with BoNT-A.”
Reviewer: Lines 40-42: please add multiple references at the end of this sentence.
Response: We have included the references provided at the end of the paragraph, at the end of this sentence. The sentence now reads: “The relative immunogenicity of each BoNT formulation is determined both by the amount of core neurotoxin protein (the active component) and the amount of accessory clostridial proteins (nontherapeutic component) [6,10].” (Lines 42-44).
Lines 42-46: please add multiple references at the end of this sentence.
Response: We have included one of the references provided at the end of the paragraph, and an additional three references to this sentence, which now reads: “Although only neutralizing antibodies (NAbs) against the core neurotoxin can interfere with the therapeutic effect and result in clinical nonresponsiveness [14], activation of the immune response by the accessory proteins, particularly the hemagglutinin-1 protein, can facilitate this unwanted NAb development [10,13,15].”(Lines 44-48).
Lines 50-52: please add multiple references at the end of this sentence.
Response: Two references have been added. The sentence now reads: “While other BoNT-As on the market contain varying amounts of accessory proteins, all accessory proteins, and other bacterial substances, such as flagellin, are removed from incobotulinumtoxinA during manufacturing [6,10].”(Lines 52-54).
Results
Lines 69-72: please add appropriate references for each scale.
Response: Please note that the reference has already been provided as part of the Materials and Methods Section, 5.1 Participants (please see Line 401 which includes reference 39), as is usual convention for manuscripts. However, we have also added respective references to the Results section (Lines 73 and 74) and footnote c of Table 1 (Line 81): “GMFCS levels [21] were…”
Lines 145-149: please add appropriate references.
Lines 151-166: please add appropriate references for each sentence.
Table 3: please add appropriate references for each result.
Response: The respective lines refer to the new work which is presented in this manuscript. Although we appreciate these suggestions, it is not possible to reference the findings/data of the new analyses. Thus, we have not made any change.
Discussion
Lines 246-258: please add appropriate references for each sentence.
Lines 283-300: please add appropriate references for each sentence.
Response: The respective lines refer to the new work which is presented in this manuscript. Although we appreciate these suggestions, it is not possible to reference the findings/data of the new analyses. Thus, we have not made any change.
Materials and Methods
Lines 393-400: please add appropriate references for each scale.
Response: Please note that references for the Ashworth Scale (Line 400) and GMFCS (Line 401) are included in the manuscript. A reference for the modified Teacher’s Drooling Scale has been added (Mier RJ, Bachrach SJ, Lakin RC, Barker T, Childs J, Moran M. Treatment of sialorrhea with glycopyrrolate: A double-blind, dose-ranging study. Arch Pediatr Adolesc Med. 2000 Dec;154(12):1214-8. doi: 10.1001/archpedi.154.12.1214. PMID: 11115305.) (Line 408).
Statistics
Comment:
Please describe this section very exactly.
Response: We have added the following: “As such, only standard descriptive analyses were performed. Continuous variables for the analysis of demographics were summarized by mean and standard deviation. For qualitative variables, absolute and percent frequencies (n, %) were displayed. Percentages were calculated using the denominator of subjects in the respective analysis set.” (Lines 514-517)